# A Qualitative Study toward Technologies for Active and Healthy Aging: A Thematic Analysis of Perspectives among Primary, Secondary, and Tertiary End Users

**DOI:** 10.3390/ijerph18147489

**Published:** 2021-07-14

**Authors:** Margherita Rampioni, Adrian Alexandru Moșoi, Lorena Rossi, Sorin-Aurel Moraru, Dan Rosenberg, Vera Stara

**Affiliations:** 1IRCCS INRCA—National Institute of Health and Science on Ageing, Innovative Models for Ageing Care and Technology, via S. Margherita 5, 60124 Ancona, Italy; m.rampioni@inrca.it (M.R.); l.rossi@inrca.it (L.R.); 2Department of Psychology, Education and Teacher Training, Transilvania University of Brasov, B-dul Eroilor 29, 500036 Brașov, Romania; adrian.mosoi@unitbv.ro; 3Department of Automatics and Information Technology, Transilvania University of Brasov, B-dul Eroilor 29, 500036 Brașov, Romania; smoraru@unitbv.ro (S.-A.M.); dan.floroian@unitbv.ro (D.R.)

**Keywords:** active aging, technology for aging well, user-centered design, thinking aloud, focus group, empowerment, coaching, personalization, security, social isolation

## Abstract

It is expected that, by 2050, people aged over 60 in 65 nations will constitute 30% of the total population. Healthy aging is at the top of the world political agenda as a possible means for hindering the collapse of care systems. How can ICT/sensing technology meet older people’s needs for active and healthy aging? This qualitative study carried out in Italy and Romania in 2020 involved 30 participants: older adults, caregivers, and stakeholders. Based on a user-centered design approach, this study aimed to understand which requirements of ICT/sensing technologies could match people’s needs of active and healthy aging. Findings highlighted that ICT/sensing technology needs to focus on six major themes: (1) learnability, (2) security, (3) independence, empowerment, and coaching values, (4) social isolation, (5) impact of habit, culture, and education variables, and (6) personalized solutions. These themes are consistent with the Active Aging framework and the factors that influence perceived usefulness and potential benefits among older adults. Consequently, this study shows how well-known, but still unresolved, issues affect the field of information and communication technologies (ICTs) to promote active and healthy aging. This suggests that the reinforcement of the public health system, especially considering the pandemic effect, requires a concrete and formidable effort from an interdisciplinary research network.

## 1. Introduction

Population aging is a global trend; there were 703 million persons aged 65 years or over in the world in 2019, and this number is projected to double to 1.5 billion in 2050 [1]. Since older adults are healthier than ever before, the concepts of active and healthy aging emerged in 1990 as the foremost policy answer to the demands of this global demographic trend [2,3]. Notwithstanding the influence of genetics, older people’s health is mostly matched with people’s physical and social environments (i.e., homes, neighborhoods, and communities), and their personal characteristics (i.e., sex, ethnicity, or socioeconomic status). Indeed, the World Health Organization (WHO) defined active aging as “the process of optimizing opportunities for health, participation, and security in order to enhance quality of life as people age” [4] for “helping people stay in charge of their own lives for as long as possible as they age and, where possible, to contribute to the economy and society” [5]. This definition opened the path to a new paradigm that considers the growing number of older individuals as a potential resource for families, communities, the economy, and society as a whole [2,4]. Even if the active aging model is commonly used to aid policy strategies in Europe, there is still no standard on how to measure it [6] or how to transfer this knowledge to other domains that could benefit from it, such as the technology field.

The rest of the paper is organized as follows: the theoretical background summarizes the main aspects related to smart technologies that support older adults to live independently in their own environment and reduce dependence on professional caregivers. Section 2 presents material and methods, which deal with the methodology related to participant inclusion, data collection, and analysis. Section 3 describes the results, containing the general data and topics relevant to the older adults, caregivers, and stakeholders. In the discussion (Section 4), the results of the study are compared with other quantitative studies in the field of ICT, as well as the limitations and implications for older people and ICTs.

### 1.1. Theoretical Background

All key components of the active aging framework can be matched with the concept of technology for aging well under the umbrella term of assistive technologies (ATs), i.e., any device, product, or equipment that helps people to perform a task they would otherwise be unable to do, or that facilitates seniors’ daily lives [7]. In fact, during the last two decades, research on technology and aging captured the interest of different scientific fields with the aim of finding cost-effective solutions to support independent living and care provision [8]. In particular, seniors prefer to age at home, and it is well known that a safe and supportive living environment is conducive to independence. It also increases the likelihood of their continuing to live in the community, either in the current home or in appropriate housing [9,10,11,12].

In addition to physical and social environments, different geographical, social, and cultural environments influence the ability of older adults to maintain mental and physical activities, as well as social interactions and personal identities [13,14]. Human beings seek to find a balance between basic needs (a safe and stable base) and higher-level needs toward exploration, stimulation, and environmental mastery, linking place attachment to the individual outcomes of autonomy and wellbeing [15]. Because technology is increasingly being integrated into every aspect of our lives, it represents an opportunity for creative solutions to promote independence and aging in place [12]. In particular, the newest smart technologies support older adults, with or without disabilities, to enable them to remain living independently in their own environment, to monitor their health status and safety circumstances, and to reduce the considerable care burden on family and/or professional caregivers [8,9,10,12,16,17]. There is a wide range of technologies that could create sustainable conditions for self-sufficient and self-determined lifestyles, for example, wearable and ambient sensors, monitoring and environmental control systems, and information and communication technologies (ICTs) [18,19].

In addition to the value of aging in place and the possibility to monitor health status and security circumstances, social isolation is a growing concern among older adults [20,21,22]. Most importantly, social isolation is a real threat to the mental and physical health of the older population, leading to depression, self-harming (e.g., drug abuse, alcoholism, and suicide) or self-neglecting behavior, a higher level of cognitive and/or physical disability, and increased mortality [23,24]. Use of ICTs can also empower older adults by engaging them in critical thinking and decision making, providing access to information and resources, and maintaining continuous contact with caregivers and friends through smartphones and/or tablets. Self-confidence and empowerment further trigger positive feelings toward themselves and control over their life and/or life satisfaction [25,26].

The challenge is to create smart living environments that are safe and secure while also preventing hazards, disability, stress, fear, and social isolation [27]. This new generation of devices may empower individuals to take care of their own health, thus promoting the central role of the person and responsibility in enabling an optimized health care ecosystem [28]. This challenge should be supported by a chain of value focused on the development of appropriate services for the promotion of active aging based on how effectively older adults want to age [3]. To this effect, the clinical field and the technological field should be allies in approaching this challenge. In particular, enabling older people to remain independent at home not only takes into account solutions for the security [29] of the surrounding environments, but also encompasses the support of people’s higher-level needs (specifically, self-empowerment and being with others) and, therefore, the entirety of the individual. In this way, higher-level needs, such as self-esteem, identity, self-actualization, and agency, are becoming increasingly important along with the most common basic physiological and safety needs [30,31]. This approach, therefore, could restore dignity and autonomy to the individual, which are primary values and represent the fundamental rights of every human being.

### 1.2. Objective

Considering the scenario presented in this section, the current study attempts to answer the following research question: Which requirements of ICT/sensing technologies could match people’s needs for active and healthy aging? The aim was to collect and analyze the perspective of older adults, caregivers, and stakeholders in the field of care and technology, on a list of devices that promote healthy aging, thereby improving individuals’ independent lives. The qualitative thematic analysis of these visions was used to identify technology macro-requirements that better fit users’ needs and expectations.

## 2. Materials and Methods

This study forms part of the European project “Safety of Elderly People and Vicinity Ensuring” (SAVE) (aal-2018-5-149-CP), aimed at providing a technology platform to support the healthy aging of older adults suffering from age-related chronic illnesses or mild cognitive issues and/or disabilities. The SAVE system is a solution whose main goal is to support end users in staying in their familiar environments for as long as possible, exercising their autonomy and self-management, and avoiding social isolation. It also supports informal caregivers, such as relatives, in providing the necessary care while maintaining their professional and private life. The SAVE system is under development following iterative cycles based on the ISO 9241-210 human-centered design for interactive systems [32]. At the time of this study, the concept of the SAVE system was supposed to be a multicomponent platform based on works with multiple smart-home and wearable sensors streamed directly to a cloud-based platform, where algorithms detect any behavioral and physiological deviation (Figure 1).

This paper outlines the main findings of the first iterative session held in Italy at the National Institute of Health and Science on Aging (INRCA) and in Romania at the “Hand in Hand” Organization from Brașov, the Direction of Social and Medical Assistance (DASM) from Timișoara, and the Social Assistance Direction (DAS) from Brașov, aimed at defining how such ICT/sensing technologies meet the intended users’ needs of active and healthy aging.

### 2.1. Inclusion Criteria

This study involved three types of participants: (a) primary end users, i.e., 65+ older adults suffering from moderate medical conditions or moderate impairments (mild dementia and/or disabilities); (b) secondary end users, i.e., informal caregivers (at the family and volunteering level); (c) tertiary end users, i.e., stakeholders including care providers, public social service, end-user organizations, and medical and nursing researchers. The inclusion and exclusion criteria of the study for the older adults were to (a) be more than 65 years old, (b) not suffer from major chronic diseases or severe disabilities (self-reported assessment), (c) retain sufficient mobility, for example, moving and maintaining body positions, handling and moving objects, moving around in the environment, and moving around using transportation (self-reported assessment), (d) receive occasional care from relatives or professional caregivers, (e) be able to complete the session, and (f) have at home a smartphone and internet access.

Exclusion criteria were to (a) be less than 65 years old, (b) suffer from major chronic disease or severe disability (self-reported assessment), (c) have insufficient mobility (self-reported assessment), (d) receive long-term care from relatives or professional caregivers, (e) be unable to complete the session, and (f) not have a smartphone and internet access. Additionally, the only inclusion criterion for caregivers and stakeholders was being in a close relationship with older adults and/or providing care for them. According to the SAVE project activities, a staff composed of two psychologists (M.R. and A.A.) contacted 30 potential participants, informed them of the project objectives, methods, and timing, and underlined that participation in the study was completely voluntary, and that they could leave the study at any time without providing any explanation. Researchers then asked participants certain screening questions in order to check their inclusion characteristics. Nobody refused to attend, and participants signed a written document giving their informed consent for the processing of their data, in accordance with the GDPR 2016 and national legislation on privacy and data protection.

### 2.2. Data Collection Methodology

Thinking aloud and focus group methods were used to collect data from the sample. The thinking-aloud method is a unique source of information on cognitive processes. The method consists of asking people to say whatever comes to mind about what they are doing, thinking, and feeling while solving a problem and analyzing the resulting verbal protocols for knowledge acquisition. This method is appropriate for studying the cognitive problems that people have in learning to use a computer system [33,34,35,36,37]. This study gathered data from older adults and family caregivers through one-to-one sessions (a researcher and a user). On the other hand, the focus group was used to collect data from stakeholders, being one of the more suited and used methodologies for collecting users’ needs, feelings, and impressions on new technology [36]. The enrolment of three types of participants (i.e., older adults, caregivers, and stakeholders) and the analysis of their perspectives ensured the study’s trustworthiness [38]. Moreover, the interpretation of data was the result of an iterative work carried out by a multidisciplinary team consisting of two psychologists (M.R., A.A.), one expert (V.S.) of user-centered design (UCD), and two engineers (L.R., S.M.). The discussions were conducted by a researcher playing the role of moderator (V.S. in Italy and S.M. in Romania), whilst another researcher (M.R. in Italy and A.A. in Romania) observed the discussions, as well as took notes and timings. The parallel and independent data analysis by the researchers minimized the research bias [39]. Since the study took place during the national quarantine imposed by the Italian government, after the exacerbation of the COVID-19 pandemic (March–May 2020), the authors were obliged to reach users remotely by phone or by means of a video conferencing platform. For the same reason, prior to the thinking-aloud session, older adults and family caregivers watched a video that first presented the project’s concept and then the list of services offered by the SAVE system. The video lasted 15 min and was accessed through the YouTube platform. After the video, a researcher phoned the user to gather data. Stakeholders involved in the focus group independently checked a file received by email containing the list of storyboards representing the concept idea of each service offered by the SAVE system (Figure 2) and then joined the discussion, led by a moderator, within the group.

Each participant reported their first reactions to the following open topics during a 30 min discussion:If and what the video or storyboards evocated in their mind (features, services, or simply suggestions to improve the system);Why they found it interesting or otherwise (which is the added value for them, the pains and gains).

### 2.3. Data Analysis Methodology

Thinking-aloud and focus group discussions were transcribed and analyzed using the framework analysis method [40,41,42]. MAXQDA software package (MAXQDA, VERBI Software GmbH, Berlin, Germany) for qualitative research was used. Researchers classified and categorized text data segments into a set of codes that were then combined under main themes. Specifically, different data segments were associated with the same code and codes were gathered under the same theme. Quotes were then sorted out and comparisons made between them [43].

## 3. Results

As shown in Table 1, our results were divided into three distinct groups; the first included 13 older adults, the second included eight caregivers, and the third comprised the nine stakeholders. The cluster of older adults included 11 women and two men aged between 67 and 90 years. The age range of caregivers was wider (between 39 and 75 years), and the group was made up of four women and four men. The stakeholders were six women and three men aged between 33 and 55. One was an end-user representative, while the others were psychologists. All the older adults lived with their spouse or alone and, except for one older adult, were able to use technology.

In accordance with the framework analysis, the thinking-aloud feedback and focus group transcriptions were associated with codes and the latter were merged into themes. Table 2 shows the themes arising from their correspondence with codes.

Six themes emerged from the thematic analysis of both thinking-aloud and focus group transcriptions: learnability (Theme 1), security (Theme 2), independence, empowerment, and coaching values (Theme 3), social isolation (Theme 4), impact of habit, culture, and education variables (Theme 5), and personalized solutions (Theme 6).

### 3.1. Older Adults’ Perspectives

The older adults group stressed the importance of learnability (Theme 1). The use of certain systems seemed to presuppose a basic knowledge of technology by seniors, as highlighted by an older woman: “*The skill to use/manage the smartphone would seem to be crucial for access to SAVE services*” (G1_P1). For this reason, older adults reported that they preferred the smartwatch to the smartphone, since the former would seem to be more helpful and of more immediate use, especially if the person is alone, wants to monitor his/her physical state, and needs fast assistance: “*I think the smartwatch may be more useful than the smartphone for an older person, especially when he/she goes out, and he/she is alone. In my opinion, older people are very stubborn; they are ashamed to feel weak and to communicate it to strangers. For this reason, in a time of need it can be useful to have the possibility of only pressing a button to receive help*” (G1_P3), “*If I have an accident, I would not be able to call for emergency help and, in this specific case, a sort of armband or smartwatch that monitors my vital signs is more useful than a smartphone*” (G1_P5), and “*If I have a memory lapse, I will also probably forget to turn to my smartphone to seek help*” (G1_P1).

Moreover, older adults reported the desire to have a system that is able to understand if a person is sick or in difficulty without doing anything, thus reducing the effort to learn, as described by the following quotation: “*Systems must be as simple as possible because technologies are always very difficult for us and we usually spend a lot of time trying to understand how technology is supposed to be used*” (G1_P4). In this regard, they pointed out the difficulty that users often have with technology and the great importance of technological assistance that could help and reassure older people: “*We need clear manuals and sometimes the help of our sons. Every activity related to instructions is really appreciated*” (G1_P4).

Another theme that developed during the sessions was security (Theme 2). One of the older people explained that “For old people, help is also needed in the event of malfunctions of the electrical, plumbing or gas installation. They need trained, professionally correct people” (G1_P 10). In this sense, the safety of the person in their own environment is fundamental by using the accessories in the house: “Every time when I leave my house I check if the gas, water, or light is on 3–4 times. I am afraid of not only the expenses, but also to not blow anything in the house and surroundings” (G1_P12). Security should have the option “to find” your own house: “I think that a device that can guide us back home when we forget the road would help us a lot. I had the misfortune to get on another bus, and, because I left the neighborhood, I no longer recognized the streets” (G1_P13). Older people feel “safety” in terms of stability, recognition of objects, tasks, and needs: “From the SAVE system, I expect to not be afraid to stay at home and forget to turn off the gas, turn off the water, so I take care of myself, but also of others” (G1_P6). Moreover, older people want to use equipment that is easy to understand without the risks of installation and use: “I saw a gas detector on TV, not expensive, but maybe this SAVE system will be able to be installed in my house, especially in the kitchen” (G1_P12).

Another strongly felt needs that emerged from this group is the need for independence, understood as an exaltation of the preservation of older people’s mental and physical capacities for as long as possible (Theme 3), as highlighted by P5: “*I am an independent person and I want to be as active as possible. So, for me, technology must boost my activity and not only offer ‘protection’ against harm or accidents or ‘control’. I hate to be controlled*”. There would seem to be a belief among older adults that technological systems are designed with a purely welfare, protection, and/or control function, without considering that certain devices could be especially useful even in the lives of older people who are still active and autonomous, as described by the following quotation: “*I do not need such a system because I am independent*” (G1_P4). Older adults, in fact, strongly stated their need to remain self-sufficient for as long as possible, and it is precisely in this sense that technology should promote activities, empowerment, and self-management. During the think aloud, this strong sentence was mentioned: “*I’m afraid I’ll die alone at home, and no one will know*” (G1_P8). Theme 4, social isolation, underlined the need to reduce this behavior: “*Since I retired, I leave the house less, I meet less often with children or friends, and sometimes I do not want to become a burden for my family*” (G1_P7). A networking issue was also emphasized: “*The connection between older people, developed through various activities promoted by the system is very important for me. For example, online movies, news, online courses, concerts,* etc. *that could be watched from home, when I cannot get out the house*” *(*G1_*P7).* Social isolation becomes a dangerous factor, if we do not detect it in time: “*You have to think of this system as a detector of inactivity, for us any activity means utility, sense of an experience. Without them, we feel without rhythm, energy, or feeling to live*” (G1_P6). Older adults also highlighted the importance of considering the link between the “habits” and “culture” variables and technology (Theme 5). Thus, it is not only older persons’ attitudes and interests in technology that should be considered, but also the environment in which they have always lived: “*I think it’s important to consider the influence that the ‘habits’ and ‘culture’ variables could have on the use of technology. In my opinion, Italy’s aging population is not very adept at technology, while the situation is quite different in other European countries. I am not very attracted to tech devices*” (G1_P2).

### 3.2. Caregivers’ Perspectives

Caregivers highlighted the importance of learnability (Theme 1). A system should also be easy for older people without having to learn new skills. Such learning is often complicated and energy-consuming not only for users but also for careers, who need to provide technological help and assistance: “*Older adults are not so confident with innovations, even if they use a smartphone or tablet; some services are too difficult for them and, in this case, it is very stressful for a caregiver providing assistance*” (G2_P17). In this sense, caregivers also speculated on the influence that “culture” and “education” variables (Theme 5) could have on the development of a positive user approach to technology. Moreover, they pointed out that, in the case of users with initial memory problems, needs become so complex that these needs would be difficult to match with technologies, in spite of the best learnable design: “*I am not sure that my mother would be able to use the smartphone and the geo-localization function to come back home if she goes out and gets lost. She would most probably be able to phone me but not remember that she has an app that could help her. In a stressful situation, persons forget everything*” (G2_P18) and “*My wife has the first symptoms of dementia and what I see is that she forgets her smartphone everywhere, as well as other important things like keys and/or glasses. My concern is that this system may not be useful for her, not because the system isn’t interesting, but just because my wife will forget it*” (G2_P19). They also stressed the need to have personalized solutions (Theme 6): “*I think that there is a need for systems that are designed around the specific needs, intentions, and barriers of users. It is better to have fewer but very well personalized services*” (G2_P17). Caregivers highlighted the importance of older people remaining independent (Theme 3) and recognized in technology a way to maintain their wellbeing, as well as cognitive and physical autonomy. For example, cognitive exercises and/or games could be useful in supporting older adults in stimulating and maintaining their skills: “*If I consider my father, I really want him to stay as active as possible and my concern is that this kind of service could lead him to becoming passive and less attentive to his daily activities*” (G2_P15) and “*My mom […] forgets things, events, appointments and so on. The system might be beneficial to her, but my wish would be for the system to support her in maintaining her capabilities, maybe with some cognitive exercises or games*” (G2_P18). Regarding Theme 2, security, the caregivers agreed: “*Using such systems have advantage to know almost all the time where the older people is, and the most important you can contact them in real time, without any obstacles. Being able to learn from the system how to return home using your intelligent system is a huge progress for personal security*” (G2_P20). The benefits are not only for the professional caregivers but also for the family caregivers, who were pleased with the safety that the system provided. Older people are involved in fewer activities compared with other populations; one of the reasons could be retirement or lack of friends: “*Such systems should have an application for involving older people in different activities (theatre, dance, concerts, going to church, visiting nature, group activities) to reduce the level of ‘social isolation’ and to prevent any risky situation for them*” (G2_P21).

### 3.3. Stakeholders’ Perspectives

With regard to the stakeholders, they also stressed the importance of learnability (Theme 1) and security (Theme 2) when introducing new devices among older adults. For example, G3_P29 said that “*Thinking of older people, we recognize the fear regarding their security in their houses, on the street, or during their walks. We never know what could happen during this time, maybe it is a good idea to have sensors that are able to detect critical situations*”. This group also underlined the importance of developing systems that promote older people’s empowerment and self-management (Theme 3) and not just provide a passive monitoring of the physical state of users: “*It is important that such devices empower users in self-management by making sure users’ self-responsibility is retained*” (G3_P23) and “*This means being active and not passive or under control*” (G3_P25). Moreover, it is also important to have a motivational coach to support users in interacting with the system, providing them with strong encouragement when they act in the right way, e.g., when they use the system daily: “*Such devices can play the role of motivational coach: it is not enough for my phone to remind me to have a walk of at least 5 km a day; I need to stay motivated to do it*” (G3_P24). The value of personalized solutions (Theme 6) was recognized by G3_P27: “*It is important to share knowledge on how technologies can help users to be active and independent in the progressive process of aging. If certain technology is not useful at this current point in time, it could become significant later when age-related conditions arise. So, for the time being, I can just keep myself informed for the near future. It is a way to acquire awareness on personalized solutions*”. Stakeholders stated their concerns about the most critical situation of “being alone at home” (Theme 4). G3_P30 summarized the general thought as follows: “*We never know what their activities are in their homes, and, without any feedback system, our options are limited. I am convinced that, if we involve them in different activities, we will reduce their fear; they will be in direct connection with them and share with us, friends, or caregivers their fears, sensations, and feelings; we will be able to adapt the system to their needs*”.

## 4. Discussion

This study was aimed at collecting and analyzing which requirements of ICT/sensing technologies could match people’s needs of active and healthy aging. Thinking-aloud discussions with older adults and caregivers, as well as a focus group with stakeholders, were held for this purpose. As shown in Figure 3, older adults, caregivers, and stakeholders agreed about six important features: learnability (Theme 1), security (Theme 2), independence, empowerment, and coaching values (Theme 3), and social isolation (Theme 4). Older adults recognized the importance of a system that not only reduces the effort to learn something new and complex, but also reduces the need to ask a relative and/or a friend for help. All three groups gave other interesting insights. They stressed the need for the system to be able to not only protect users’ vulnerabilities and security, but also to support them in maintaining their abilities and to make them feel as active and determined as possible. Older adults significantly underlined the impact of habit, culture, and education variables (Theme 5), whereas the importance of having personalized solutions (Theme 6) was predominant among caregivers. The themes that emerged are consistent with the Active Aging framework (Themes 3, 4 and 5) and the factors that influence perceived usefulness, security, and potential benefits (Themes 1, 2 and 6) among older adults [5,44,45,46,47].

With regard to Theme 1, results showed how important it is for older people to be able to learn and use technological devices to reach and achieve some desirable outcomes with as little cognitive effort as possible. The design of most electronic household appliances, such as the buttons on control panels and font sizes, colors, and touch screens, are not age-friendly, and the instructions are complicated, without the full details, and hard to follow. For this reason, older adults may feel confused or frustrated when using technology [48]. This use can be influenced by how well such technology meets older adults’ daily needs and physical problems [49]. Users are more open to the use of technology if they perceive its usefulness in dealing with everyday life. For instance, for individuals worried about taking medication exactly as prescribed by the doctor, a reminder technology, e.g., medication management/dispenser, would be expected to satisfy their needs and to be adopted promptly [50]. Peek et al. [49] showed that older adults’ acceptance of technology was associated with their attitudes toward the benefits of technology, as well as the perceived consequences, personal proficiency, perceived need, and willingness to use it [45,49,51,52]. Therefore, it is necessary to provide older people with both technical and emotional support for the use of technology [53,54]. Older adults seemed to be concerned about safety and privacy threats while using the Internet [55,56,57,58,59]. Moreover, when their relatives or friends were showing them how to use the appliances, they felt anxious about making mistakes, not being properly focused, or quickly forgetting the instructions [60,61]. Since lower self-efficacy and higher computer anxiety predict lower use of technology, the learning perspective in the development and deployment of new technologies should play a major role so as to avoid the exclusion of older users [62,63]. According to del Barrio et al. [64], aging is influenced by environmental, economic, cultural, and social conditions of a concrete context, which provides opportunities and resources or creates barriers for older people. Several determinants both at a population and at an individual level could have an influence on Active Aging and contribute to wellbeing. All of these are influenced by two transversal factors: culture and gender. Personal determinants refer to the individual biological, psychological, and behavioral conditions of aging people, whereas contextual determinants include socioeconomic, sociopolitical, and environmental factors affecting the environment in which people age.

As emerged from Theme 2, all three groups agreed that technological devices should provide a sense of confidence and security for older adults, which would enable aging in place. Starting from the concept of a smart home, i.e., a residence equipped with technologies that include sensors, wired and wireless networks, and intelligent systems, over the past decade, smart home technology has increasingly targeted people having reduced capabilities due to age or disability. It was initially focused on increased security, convenience, and energy savings, whereas, recently, the scope of its use has gradually varied to enhance the overall quality of life. Smart homes increase domestic comfort, convenience, security, and leisure, as well as reduce energy use through optimized home energy management [65,66]. Lee and Kim [67] stated that safety, health, sustainability, and convenience are suggested as the four major goals in the development of an intelligent living space policy. The older adults referred to security and safety as the top value. According to Turjamaa et al. [68], older people reported that smart homes improved their sense of security at home (especially if they lived far away from their relatives), as well as their quality of daily life and activities, and provided them with information about the care they could receive. Therefore, the smart home solutions would be designed to help older people carry out everyday activities and lead healthier and more fulfilled lives, by improving their physical safety and social communication.

With regard to personal determinants, as discussed in Theme 3, older adults, caregivers, and stakeholders stressed the need for a technological system capable of protecting the vulnerabilities of users, as well as to support them in maintaining their abilities and to make them feel as active and independent as possible. Stakeholders also underlined the value of empowerment and self-management to promote wellbeing. Several studies [15,36,60,69,70,71,72,73,74] have claimed that older people want to remain independent for as long as possible. According to Mannheim et al. [73], autonomy concerns a person’s right to independently make their own decisions about life. In relation to older adults, this is highly associated with decision making concerning care, housing, social activities, and even the end of life. In order to lead independent and healthy lives in their own homes, people should be able to perform a wide range of everyday tasks including self-maintenance (ADLs) and instrumental activities of daily living (IADLs). In addition to sustaining a situation of independence and autonomy, personal growth and development are important aspects of a meaningful life. Personal growth activities, including the willingness to accept new challenges and to engage in lifelong learning, are primarily aimed at personal enrichment, self-fulfillment, and pleasure. In addition, they imply adjustment to changes, such as keeping up with technological and communication developments (e.g., the Internet, smartphone, and tablet) [71]. Indeed, Ahn et al. [74] underscored the importance of self-esteem as a significant factor in psychological wellbeing in later life. Self-esteem has been found to be positively associated with happiness, perception of physical health, and longevity. Older adults did not view themselves as an archetypal “old” person but rather, showed a strong desire to preserve and portray an identity associated with self-reliance, competence, and independence. Moreover, the desire to remain independent stemmed from their wish to not be perceived as a burden to family, friends, or society in general. Along with their desire to preserve independence, there was that of maintaining control in their daily lives. The latter resulted in older adults’ rejection of helpful and beneficial technologies which they perceived as stigmatizing or reinforcing an image of being “old”; they, therefore, seemed to prefer the technologies they could be “in charge of” [72]. For this reason, in relation to older adults, digital technology (DT) holds promise to improve wellbeing and healthcare, as well as to support aging in place in a safe and independent environment. In today’s rapid acceleration of a digitized environment, the implementation of DT use may assist older people to live in their own home and facilitate their engagement in everyday tasks, helping to maintain their independence and increasing their control over the world around them [69,72,73].

With regard to Theme 4, results showed that the contribution of technological systems toward improving the quality of life for older adults should also focus on social and fun aspects. Home features hardwired with smart communication technology can be used to encourage their social connectedness and prevent those living independently from feeling isolated. Previous studies [20,21,22,24] highlighted the benefit of a support network for frail older people or their caregivers; this could be valuable in preventing institutionalization and disability. Moreover, personal interaction is a necessary component of care-giving [65]. According to Lee and Kim [67], older adults’ active participation in social activities and the establishment of their sense of belonging as a social member have important effects on successful aging in place. Therefore, DT would be an effective way to overcome social isolation among older people through connecting to the outside world, gaining social support, engaging in activities of interests, and boosting self-confidence.

With regard to contextual determinants, as discussed in Theme 5, older adults and caregivers suggested looking carefully into the influence of the “habits”, “culture”, and “education” variables on technology acceptance. This involves not only considering older people’s attitudes and interests in technology, but also their surroundings [75,76,77]. Some studies [36,71] described education, culture, and income levels as predictors of technological adoption and use. According to Quan-Haase et al. [60], age-related factors, beyond income, education, and gender, affect older adults, hindering their ability to take advantage of digital technology. For example, low digital literacy could hinder older people’s use of digital media, perhaps because most did not grow up with it and had to learn skills later in life [78]. Older adults believed that updating their digital skills and habits required committing significant amounts of time and energy to the challenge, and they viewed this as too costly. They felt they simply did not have the time to spare. With their days filled with other activities, they wanted to devote themselves to such things [60,79]. What has been described so far is closely related to the concept of “aging in place” that represents the desire expressed by older people to continue living within the community, with some level of independence, rather than in residential care [15,64]. Housing and neighborhood satisfaction have been used as good indicators of environmental and overall wellbeing for older adults [74]. Moreover, maintaining older people’s good mental abilities and social relationships turns out to be a way to enhance social connectedness, i.e., the feeling of being connected to other individuals and communities, as well as quality of life [15,70]. In fact, older adults often choose to live near a religious or other social organization to maximize opportunities for social interactions and enhance life satisfaction [71]. Social connectedness is important for older people, not only because it provides them with the feeling that they are valuable to others, but also because a lack of social connectedness could threaten their physical and mental health [74].

In closing, another significant aspect that emerged from the results is the push for personalization (Theme 6), since a “one size fits all” approach is not suited to a diverse set of users [80]. The need is to adapt content and functionalities to the aims, behaviors, preferences, context, and lifestyle of the intended user. This means “changing the system functionalities, interface, content, or distinctiveness to increase its personal relevance to an individual or a category of individuals” [81]. For example, personalization is primarily used for tailoring content through feedback, daily health reports, alerts, warnings, and recommendations; nevertheless, this recognized added value is not yet considered a distinctive design factor [82].

### 4.1. Limitation

Despite the benefits of collecting and comparing these different perspectives, the specific national centrality on Italy and Romania, as well as gender disparities in the sample size, could be seen as bias and significant limitations that do not allow for the generalization of results. Furthermore, the sole use of qualitative methods is also a limitation, whereas a mixed-methods approach could have guaranteed a broad understanding of thoughts and emotions toward the use of ICT/sensing technology to promote older adults’ healthy aging. Focus groups are the qualitative method most used in studies on the use of technology to grasp the perspectives of respondents regarding the usability and acceptability of the artefacts [83]. Since this qualitative method is aimed at understanding the perspective, meanings, and experience of respondents and not at providing the results statistical representativeness [84], the involvement of a limited number of respondents ensured the achievement of this objective by allowing a better moderation of the group discussion and, therefore, the collection of better-quality results [85]. There are no guidelines for deciding the number of groups, and, in the literature, the general rule seems to be that the sample size should refer to the number of groups and not the total number of participants in a study [86,87,88,89]. In light of the above, what makes a focus group effective is not the number of participants but their ability to bring new and interesting content. Therefore, the sample size in this study referred to the “theoretical saturation” concept by Glaser et al. [90] and recalled by Strauss and Corbin [91], according to which data have been gathered until new and relevant information is collected about the topic of the focus group.

Moreover, this study was performed in March–May 2020 in between the quarantine restrictions imposed by the Italian and Romanian governments after the spread of COVID-19. The use of physical distancing methods to reach users by phone or a video conferencing platform could have influenced the sample’s natural responses. The recent coronavirus pandemic has forced many governments to enforce physical distancing measures, which has unfortunately become social distancing, in order to contain the infection. This has introduced further complications to health and wellbeing, especially among older adults [92]. In these emergency situations, ICTs mitigated the consequences of physical distancing by helping older people to maintain relationships with their relatives and friends and to practice physical activities at home [93]. Indeed, public policies supporting digital and health literacy and the adoption of simple, user-friendly, and inexpensive digital solutions that may promote healthy lifestyles among all older adults would be welcome, even for those with fewer economic opportunities and a medium–low level of education.

### 4.2. Implication for Users and ICT

The pandemic opened great opportunities and implications for the design, development, and use of information systems and technologies in the public health system, especially in response to the changing needs of the aging population worldwide. However, health and care systems are facing increasing challenges today. Significant improvements are urgently needed to reach high-quality, efficient, accessible, health-promoting, people-centered, resilient, health and care systems especially for older adults. For example, this study identified these six pillars on which action can be taken from the different points of view of primary, secondary, and tertiary end users. Older adults, caregivers, and stakeholders identified six important features that match the users’ need related to learnability (Theme 1), security (Theme 2), independence, empowerment, and coaching values (Theme 3), and social isolation (Theme 4). The key solution could, therefore, be to enable close collaboration between interdisciplinary research networks such as medical scientists, technology developers, stakeholders, and older adults, as well as their formal and informal caregivers. This could facilitate the way forward toward building a common vision to set shareable objectives and to foresee the expected impacts as a matter of urgency in this time of societal transformation.

## 5. Conclusions

By analyzing data emerging from thinking-aloud sessions with older adults and their caregivers, as well as the focus group with stakeholders in technologies for aging well, this study discussed six important features that ICT/sensing technologies could match to support the needs of active and healthy aging: the challenge to assure the learnability of technologies for active and healthy aging (Theme 1), the significant impact that security worries have in the perspective of older adults and their caregivers (Theme 2), the added value to promote independence, empowerment, and coaching to maintain effectively an active aging perspective among older adults (Theme 3), and the avoidance of social isolation (Theme 4), while always taking into consideration the value of habit, culture, and education variables (Theme 5), as well as the request for personalized solutions (Theme 6). So far, these are all well-known, but still unresolved, issues affecting the field of ITCs to promote active and healthy aging. This suggests that the reinforcement of the public health system, especially considering the pandemic effect, requires a concrete and formidable effort from an interdisciplinary research network made up of medical scientists, technology developers, and social and business innovators, as well as the direct engagement of older adults and their formal and informal caregivers. The challenge facing demographic change is pushing the innovation enabled by digital solutions; however, there is a concrete need to understand the triggers and enablers that support the deployment of digital technologies.

## Figures and Tables

**Figure 1 ijerph-18-07489-f001:**
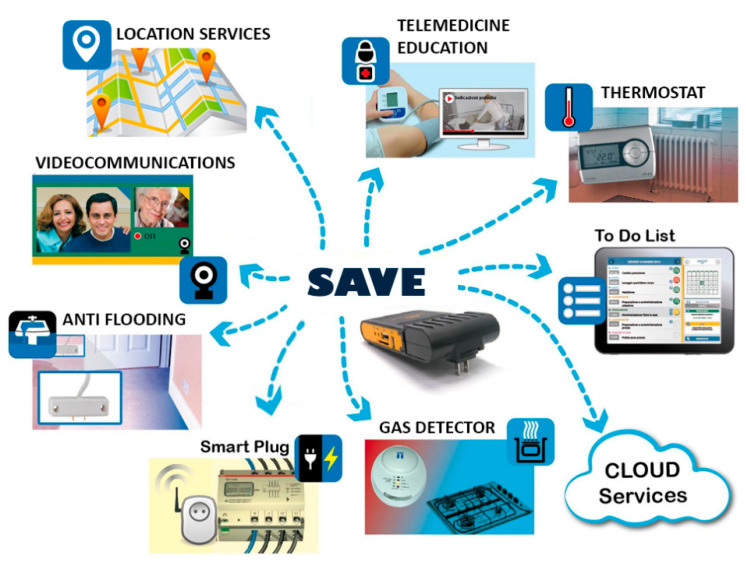
The original concept of the SAVE architecture based on multiple smart-home and wearable sensors streamed to a cloud-based platform.

**Figure 2 ijerph-18-07489-f002:**
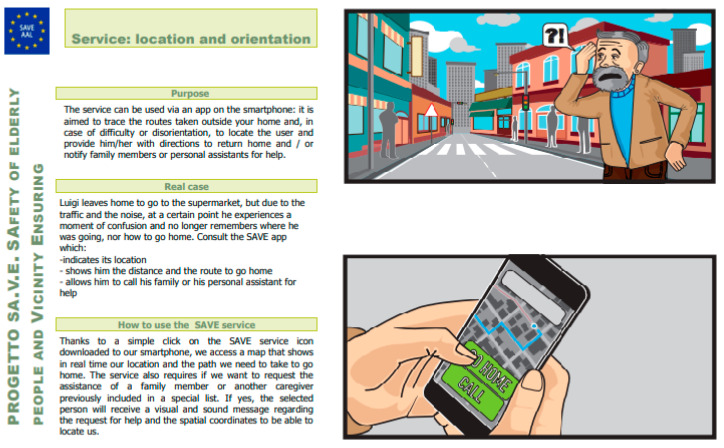
An example of a storyboard checked by stakeholders prior to the focus group’s engagement. The storyboard shows how the user would interact with the SAVE system.

**Figure 3 ijerph-18-07489-f003:**
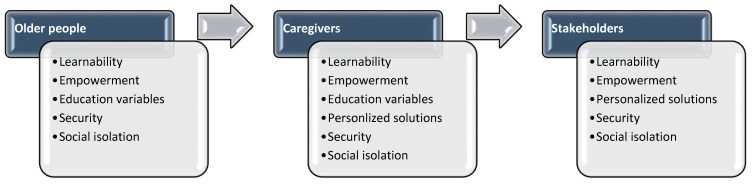
The six themes emerged from the thematic analysis in the three groups.

**Table 1 ijerph-18-07489-t001:** Participants’ characteristics.

Participants (Group_Code Participant)	N	Age (Mean; SD)	Gender	Living Condition	Using IT Devices	Nation
Older adults (G1_ P1–P13)	13	78.31 (6.62)	11 women 2 men	8 living alone 5 with spouse	87%	5 from Italy 8 from Romania
Caregivers (G2_P14–P21)	8	51.8 (11.06)	4 women 4 men	1 with spouse 5 with mother 2 with father	100%	5 from Italy 3 from Romania
Stakeholders (G3_P22–P30)	9	43.67 (17.73)	6 women 3 men	-	100%	5 from Italy 4 from Romania

**Table 2 ijerph-18-07489-t002:** Themes arising from the thematic analysis per group of respondents.

	Themes	Older Adults	Caregivers	Stakeholders	Sum
1	Learnability	13	6	3	22
2	Security	5	3	2	10
3	Independence, empowerment, and coaching values	3	4	3	9
4	Social isolation	3	1	2	6
5	Impact of habit, culture, and education variables	3	1	-	4
6	Personalized solutions	-	3	1	4
	Total Sum	27	18	11	55

## Data Availability

Data and materials are available on request from the corresponding author.

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
