# Peer review of "A Qualitative Study toward Technologies for Active and Healthy Aging: A Thematic Analysis of Perspectives among Primary, Secondary, and Tertiary End Users"

_ijerph, 2021, doi:10.3390/ijerph18147489_

Round 1
Reviewer 1 Report
The changes made to the paper have improved the quality and the information is much clearer. The article is interesting and I think readers will enjoy the content.
My main concern with the article is still that the survey groups are relatively small. There seems to be 30 stakeholders in total. I suggest that references are included to justify the small use case group. Perhaps within section 4.1 Limitations, an example (or examples) of other works with small sample sizes could be emphasised as justification for presentation of a qualitative approach with a smaller group. Otherwise, the paper is high quality and relevant to the journal.
Author Response
Dear Reviewer, thank you for this additional suggestion and the time spent to improve our manuscript. We added the following sentences in the Limitation: "the sole use of qualitative methods is also a limitation, whereas a mixed-methods approach could have guaranteed a broad understanding of thoughts and emotions towards the use of ICT/sensing technology to promote older adults’ healthy ageing. By the way, focus groups are the qualitative method most used in studies on the use of technology to grasp the perspectives of respondents regarding the usability and acceptability of the artefacts [83]. Since this qualitative method is aimed at understanding the perspective, meanings, and experience of respondents and not at providing the results statistical representativeness [84], the involvement of a limited number of respondents ensures the achievement of this objective by allowing a better moderation of the group discussion and therefore the collection of better-quality results [85]. There are no guidelines for deciding the number of groups and in the literature seems to be the general rule that the sample size should refer to the number of groups and not the total number of participants in a study [86-89]. In light of the above, what makes a focus group effective is not the number of participants but their ability to bring new and interesting content. Therefore, the sample size in this study referred to the “theoretical saturation” concept by Glaser et al. [90] and recalled by Strauss and Corbin [91], according to which data have been gathered until new and relevant information was collected about the topic of the focus group."
Methodological references are included to justify the small use case group.
Best regards. The authors
Reviewer 2 Report
Thanks to the authors for addressing my comments. I have no further concerns.
Author Response
Thank you so much
This manuscript is a resubmission of an earlier submission. The following is a list of the peer review reports and author responses from that submission.
Round 1
Reviewer 1 Report
Dear Authors:
First of all, I would like to thank you for the opportunity to review the paper entitled: A qualitative study towards technologies for active and healthy aging: what do older adults, family caregivers and experts really want? In this research, a qualitative analysis is carried out on the application of detection technology for active and healthy aging, analyzing its application from three aspects: Older adults, caregivers and experts. I consider that the idea of ​​the document is adequate but it lacks sample weight to endorse the statements. The sample size used is very small: older adults (n = 5), caregivers (n = 5 and experts (n = 5). Thematic areas (Learning, Independence, empowerment and coaching values, Impact of habit, culture and education variables, Personalized solutions).
I encourage the authors to involve more agents in this research and thus the findings and conclusions will be much more enriching.
Reviewer 2 Report
Overall the paper has an interesting and relevant topic is very relevant and the written English level is good. However, the main results are poised on presenting the response to a discussion with focus groups. I would suggest that the current evaluation of the responses is not at an appropriate level of technical contribution to be accepted to a journal.
A more technical analysis of the focus group data collection is required. For example a quantitative social science approach could be adopted using univariate/bivariate analysis. The approached used at the moment is qualitative, however technical approaches for a qualitative method are also not used, discussed or validated.
The main research question 'How can ICT/sensing technologies meet people’s needs of active and healthy aging' is not addressed directly in the discussion.
The term 'experts' is frequently used in the manuscript - but it needs defining. 'Experts' is a rather ambiguous concept and should either be more clearly defined (beyond the 'experts in the field of care and technology') or replaced with a more appropriate term.
'IT expertise' is not quantified, what is the measure for having IT expertise?
In table 1, the experts are all female, but the other groups are mixed gender. Does gender or gender balance have implications on the response? Many considerations such as this are left open-ended and not addressed.
For example, the Conclusions are quite short in comparison with the other sections and could be expanded.
Much of Section 4 Discussion would be better placed as a background section on related works earlier in the paper.
Justification for the group size of the participants is also not provided. The groups are very small, 5 in each, which means your results may be subject to bias or misleading because an appropriate sample size it not reached or justified.
Reviewer 3 Report
"A qualitative study toward technologies for active and healthy ageing: what do older adults, family caregivers and experts really want? " is an important and interesting manuscript. However, the current version of the manuscript has some limitations, and if authors can address the same and make the suggested changes, it would benefit the manuscript.
Major comments:
- Please provide details and a flow chart for inclusion criteria. Provide details on sampling, i.e., selection of the study subjects. Like how many were excluded before finalizing the 15 subjects? Also, provide details on sampling type? Is it convenience sampling?
- Provide details on how the authors confirmed or measured the included older adults who were suffering from moderate medical conditions/impairments.
- The authors have mentioned addressing researcher bias; however, authors have not provided information on kappa statistic.
Minor comments:
- Please provide the details of abbreviation of ICT in the abstract as it was used for the first time.
Reviewer 4 Report
The paper is good both in content and in the writing about AAL for the elderly, identifying four fundamental pillars on which action can be taken from three different points of view.
The paper identifies the problem and although, as the authors say, it does not solve it, at least it details the experience of a trial with few people (15) who are close to the caregivers in the midst of the confinements due to the COVID pandemic, which is precisely the age group most vulnerable to it. However, one of the positive things that this confinement surely leaves us with is an obligatory digitalisation of the homes in which the elderly have been active participants in the use of tools that perhaps in normal conditions they would not have used.
I could only recommend to the authors to use a single style recommended by the journal editor.
As most of the expressions are in American spelling it would be to keep the same with one of the words that is continually repeated throughout the text "aging" when the American spelling would be “ageing”.